# Tuberculosis-Related Knowledge, Attitudes, and Practices Among Healthcare Workers in Atlantic Canada: A Descriptive Study

**DOI:** 10.3390/tropicalmed10080214

**Published:** 2025-07-30

**Authors:** Harold Joonkeun Oh, Moira A. Law, Isdore Chola Shamputa

**Affiliations:** 1Faculty of Medicine, Dalhousie Medicine New Brunswick, Saint John, NB E2L 4L5, Canada; 2Department of Psychology, University of New Brunswick, Saint John, NB E2L 4L5, Canada; moira.law@smu.ca; 3Department of Nursing and Health Sciences, University of New Brunswick, Saint John, NB E2L 4L5, Canada; chola.shamputa@unb.ca

**Keywords:** knowledge, attitudes, practices, tuberculosis, infection control, preparedness

## Abstract

Introduction: Despite the key role of healthcare workers (HCWs) in tuberculosis (TB) prevention and control, there is a lack of regional data on their knowledge, attitudes, and practices (KAPs) regarding the disease in Atlantic Canada. Objectives: To assess the KAPs of HCWs and identify targets for educational interventions to enhance TB care and control. Methods: A cross-sectional study was conducted among HCWs in Atlantic Canada aged ≥19 years from October 2023 to February 2024. Participants were recruited via multiple channels such as social media, collegiate email lists, and snowball sampling. Survey data were collected using an online platform and analyzed using IBM SPSS Statistics v29. KAPs were assessed using Likert-type scales and internal consistency was evaluated using Cronbach’s alpha. Results: A total of 157 HCWs participated in this study (age range: 19 to 69 years); most were women (*n* = 145, 92%), born in Canada (*n* = 134, 85.4%), with nearly three-quarters (*n* = 115, 73.2%) who had never lived outside of Canada. Study participants demonstrated moderately high knowledge *(M =* 29.32, *SD* = 3.25) and positive attitudes (*M* = 3.87, *SD* = 0.37) towards TB and strong practices (*M* = 4.24, *SD* = 0.69) in TB care; however, gaps were identified in HCW abilities to recognize less common TB symptoms (e.g., rash and nausea), as well as inconsistent practices in ventilation and pre-treatment initiation. Internal consistency analysis indicated suboptimal reliability across all three KAP domains, with Cronbach’s alpha values falling below 0.7, thwarting further planned analyses. Conclusions: This study found overall moderate-to-strong TB-related KAPs among HCWs in Atlantic Canada; however, critical gaps in knowledge and practice were noted. This new information can now guide future educational initiatives and targeted training to enhance TB preparedness and ensure equitable care for patients in the region.

## 1. Introduction

Tuberculosis (TB), caused by *Mycobacterium tuberculosis*, remains a global health challenge that was responsible for approximately 10.8 million cases and 1.25 million deaths in 2023 worldwide [1]. Although Canada has one of the lowest TB incidence rates in the world, with the overall incidence rate fluctuating between 4.6 and 5.1 cases per 100,000 population in the last decade, pronounced disparities in certain populations have been described [2]. For instance, systemic health disadvantages and socioeconomic disparities (e.g., overcrowded housing and barriers to healthcare access) contribute to disproportionately high TB burden among Indigenous peoples, with Inuit Nunangat experiencing rates 300 times higher than the Canadian-born, non-Indigenous population [3,4]. Similarly, foreign-born individuals now account for approximately 77% of all TB cases in Canada [5], and with a high influx of immigrants, i.e., 1.3 million immigrants arriving from 2016 to 2021, there is a need for the healthcare system to prepare for potential increases in cases of transmissible diseases, such as TB, that may be endemic in immigrant-source countries but not common in Canada. This is important in order to prevent, control, and provide adequate, just, and equitable care for all [6].

Healthcare workers (HCWs) are vital to infection prevention and control (IPC), particularly in identifying and managing transmissible diseases like TB. Their knowledge, attitudes, and practices (KAPs) all directly impact IPC effectiveness. Up-to-date knowledge of TB enables HCWs to implement appropriate patient care practices, reduce nosocomial infections, and decrease their own occupational health risks even in low-incidence settings [7,8]. Attitudes are also important to assess as they directly influence evidence-based practice (EBP) [9], affecting care quality and potentially perpetuating stigma when fear or prejudice exists [10]. Despite being a traditionally low-incidence region, albeit with increasing TB cases in the last decade, Atlantic Canada lacks published data on TB-related KAPs among HCWs, limiting the ability to tailor local interventions.

The knowledge–attitude–practice (KAP) theory is a widely used framework in health education and behavioral interventions which asserts that knowledge is the foundation for forming positive attitudes, which then drive behavioral changes, ultimately leading to improved health outcomes. In essence, the KAP theory posits that individuals acquire knowledge, develop attitudes based on that knowledge, and then translate those attitudes into actions [11]. Although TB KAP studies are well-documented in high-incidence countries, research in low-incidence settings like Canada remains limited [12]. To our knowledge, no studies have investigated TB KAP among HCWs in Atlantic Canada, a region that includes four provinces: New Brunswick (NB), Newfoundland and Labrador (NL), Nova Scotia (NS), and Prince Edward Island (PEI), each with distinct epidemiological and demographic characteristics. This study seeks to address this gap by exploring the KAP of Atlantic Canadian HCWs and identifying opportunities for targeted educational interventions to enhance TB care and IPC.

## 2. Methods

### 2.1. Study Design, Setting, and Population

This cross-sectional study was conducted among HCWs in Atlantic Canada from 12 October 2023 to 6 February 2024. An estimated sample size (*N* = 138) was generated via G*Power software (version 3.1.9.7; Heinrich-Heine-Universität Düsseldorf, Düsseldorf, Germany)., with the alpha level set at 0.05 to control for Type I errors, the power set at 0.95 to control for Type II errors, and an anticipated medium effect size of 0.3. Another 20% (n = 28) of participants were gathered to ensure quality data for the planned analyses.

Four broad research questions were formulated: (1) What is the level of tuberculosis-related knowledge amongst Atlantic Canadian HCWs? (2) What is the level of tuberculosis-related attitudes amongst Atlantic Canadian HCWs? (3) What is the level of tuberculosis-related practices amongst Atlantic Canadian HCWs? (4) What relationship exists between the knowledge, attitudes, and practices amongst Atlantic Canadian HCWs? Further, six separate hypotheses based on the published literature were generated to examine associations between the level of knowledge, attitude, and/or practice of Atlantic Canadian HCWs and their sociodemographic characteristics, i.e., age, gender, marital status, recent TB training, education, and main source of income [13,14,15,16,17]. The two inclusion criteria for participation in the current study were being 19 years or older and currently practicing as an HCW in Atlantic Canada.

### 2.2. Survey Instrument and Scoring System

Data were collected using a questionnaire administered via Qualtrics, an online survey platform [18]. Adapted from the World Health Organization survey guidelines [19] and similar studies [14,17,20], the questionnaire was tailored to Atlantic Canadian HCWs with minor wording adjustments. Two TB-related knowledge questions asking about the perceived seriousness of the disease and the seriousness of TB in their country/region included a fourth answer option titled ‘not serious’ to allow the responses to be analyzed as a Likert-type scale question. Secondly, one TB-related attitude question asking, “Do you feel TB is a major threat to public health in Gabon” had the word ‘Gabon’ changed to ‘Canada’.

The questionnaire was piloted on five HCWs to ensure clarity and content validity. Based on the feedback, revisions such as personalizing questions, changing the age item from actual age to age range, changing the residence item from country to province, removing redundant questions, and improving the overall flow to enhance respondent understanding and survey usability were made, though their responses were excluded from the study.

Recruitment was conducted via email and intra-healthcare networks. Snowball sampling is a respected recruitment strategy known for its cost-effectiveness and time efficiency. It was also deemed an appropriate strategy for recruiting HCWs as they are not always an easily accessible population and would respect and trust colleagues, referring them to participate in the current study. Advertisements included the study objective and a survey link. To encourage participation, respondents could enter a random draw for one of five CAD 100 electronic Visa gift cards.

Scoring respondents’ answers on the TB-related knowledge questions involved awarding one point for each correct response, with no point awarded for incorrect or unanswered questions. The total achievable knowledge score was 44 points. Attitudes and beliefs about TB in healthcare were assessed using a 5-point Likert scale, with five indicating a positive/accepting attitude and one indicating a negative/prejudicial attitude. The 19 attitude-specific questions yielded a mean score representing each participant’s overall attitude, with scores closer to 5 reflecting positive attitudes, e.g., low stigma, and scores near 1 indicating negative attitudes, e.g., high stigma. Similarly, TB-related practices were assessed using 13 practice-specific questions using a 5-point Likert scale, with the mean score representing overall practice appropriateness. Scores closer to five indicated appropriate TB-related practice, while scores near one reflected poor practice.

### 2.3. Statistical Analysis

Data collected via Qualtrics was analyzed using IBM SPSS Statistics (version 29; Armonk, NY, USA). Non-respondents and unqualified individuals, i.e., not currently practicing in an Atlantic province, were excluded, yielding a final sample size of *n* = 157.

Frequency distributions were calculated for general demographics (e.g., age, sex, ethnicity, countries previously lived in) and professional demographics (e.g., healthcare role, institution of training, province of practice). Descriptive statistics summarized the data. For each main variable—knowledge, attitude, and practice—a measure of central tendency (mean; *M*), dispersion (standard deviation; *SD*), and range (minimum and maximum) was determined. Knowledge scores were reported as both the total sum (out of 44) and individual item scores (0–1 per item). Attitude and practice scores used mean values to reflect overall trends. Likert scale data were interpreted based on Sozen & Guven’s work [21]: strongly disagree (1.00–1.80), disagree (1.81–2.60), neutral/mixed (2.61–3.40), agree (3.41–4.20), and strongly agree (4.21–5.00). Internal consistency of the KAP subscales was assessed using Cronbach’s alpha. All subscales demonstrated unacceptable levels of internal consistency, which was likely due to the low number of items in some subscales, the heterogeneity of responses, and possible multidimensionality as items were drawn from a variety of scales based on similar work [13,14,15,16,17]. The TB-related knowledge subscales regarding symptoms (Cronbach’s alpha = 0.586, n = 14), transmission (Cronbach’s alpha = 0.671, n = 8), prevention (Cronbach’s alpha = 0.502, n = 7), affected demographics (Cronbach’s alpha = 0.502, n = 5), and cure (Cronbach’s alpha = 0.290, n = 5), as well as the attitude (Cronbach’s alpha = 0.316, n = 19) and practice (Cronbach’s alpha = 0.683, n = 23) subscales, all demonstrated unacceptable reliability estimates barring the planned analyses involving inferential statistics.

### 2.4. Ethics Considerations and Confidentiality

Participants received a study description integrated with the consent form and could access the survey only after selecting “provide consent”. To ensure eligibility, only those aged 19 years or older could proceed. Participants were free to withdraw at any time without any penalty. All responses were anonymous, with no personally identifiable information or internet protocol (IP) addresses collected or recorded. The study was approved by the Horizon Health Network (RS #: 2023-3233), the University of New Brunswick (REB #2023-061), and the Saint Mary’s University (REB #25-075) research ethics boards.

## 3. Results

### 3.1. Sociodemographic Characteristics

Survey data were collected from 157 HCWs currently employed in Atlantic Canada. Most respondents were female (*n* = 145, 92%), while 11 identified as male. One participant opted not to disclose their sex. There was a broad range of age groups noted across participants, ranging from 19 to 29 years (*n* = 19, 12%) to 60 to 69 years (*n* = 14, 9%), with the largest group of respondents aged 30 to 39 years (*n* = 44, 28%) (see Table 1).

Most participants identified as White (*n* = 139, 88%), followed by Southeast Asian (*n* = 8, 5%), Black/African (*n* = 4, 3%), South Asian (*n* = 2, 1.3%), Indigenous (*n* = 2, 1.3%), and other (*n* = 2, 1.3%). “Other” included Arabic and mixed ethnicity. No participants identified as East Asian or Pacific Islander (see Table 1).

Most participants were born in Canada (*n* = 134, 85.4%), while the remaining 23 respondents (14.6%) were born abroad, including the Philippines (*n* = 4, 2.5%), Nigeria (*n* = 2, <2%), and England (*n* = 2, <2%). Brazil, Egypt, Germany, Libya, Puerto Rico, Serbia, Sri Lanka, Uganda, the USA, and Yugoslavia were all represented by one participant each. Similarly, most HCWs (*n* = 115, 73.2%) had never lived outside Canada, while 39 (24.8%) had lived abroad, with the USA (*n* = 10, 6.3%), the Philippines (*n* = 5, 3.2%), and England (*n* = 4, 2.5%) being the most common countries cited.

During the study period, 156 respondents (99.4%) resided in Canada, predominantly in the provinces of NB (*n* =122, 77.2%) and NS (*n* = 21, 13.3%). There were no participants living in NL, and only one respondent residing in PEI. Interestingly, there were respondents living outside the Atlantic provinces, i.e., Alberta (*n* = 2) and Ontario (*n* = 2), commuting to work as HCWs in Atlantic Canada. Likewise, eleven respondents did not specify their province of residence but reported working in Atlantic Canada, thereby still meeting the inclusion criteria for this study.

### 3.2. Professional Demographics

The professional experience of participants in the current study varied substantially, with a range from 1 month to 576 months (48 years). The mean level of experience was 229.48 months (approximately 19.1 years, *SD* = 142.82 months or 11.9 years). Most participants reported training at community colleges (*n* = 89, 57%); the remaining reported training at universities (*n* = 62, 39%). Three participants did not report their level of training, and another three indicated training abroad without providing further details. Most HCWs were registered nurses (RNs, *n* = 105, 66%), followed by licensed practical nurses (LPNs, *n* = 18, 11%), registered respiratory therapists (RRTs, *n* = 18, 11%), supervisors/educators (*n* = 13, 8%), occupational therapists (OTs, *n* = 2, 1%), and personal support workers (PSWs, *n* = 1, <1%) (see Table 2).

Most respondents were currently working in hospitals (*n* = 119, 76%), followed by long-term care facilities (*n* = 16, 10%) and community/clinic settings (*n* = 11, 7%). At the time of the current study, most participants were working in NB (*n* = 132), followed by NS (*n* = 21), PEI (*n* = 1), NL (*n* = 1), and multiple locations (*n* = 2) (see Table 2). Fifty-one participants (32%) had previously worked in other provinces, and 23 (15%) participants had worked abroad, including Syria, the USA, and Nigeria. Examining the personal experience of participants with TB, we found most (*n* = 60, 38.22%) had previously received TB-specific training, five participants (3.18%) had personally suffered from TB, and a surprisingly large number (*n* = 66, 42.04%) for a low-incidence region knew of someone who has or had TB in the past.

### 3.3. Knowledge of HCWs About TB

A total of 157 participants completed the TB knowledge questionnaire. Correct answers were assigned a score of one, and incorrect answers were assigned a value of zero, with a total correct score generated by adding the correct number of responses. Forty-four TB-related knowledge questions were asked, with a mean correct knowledge score found to be 29.32 (*SD* = 3.25) and a range of 21 to 38 correct answers given. More generally stated, no participant knew all the correct answers, and at least one participant knew less than half of the correct answers.

#### 3.3.1. Signs and Symptoms

Generally, the recognition of TB signs and symptoms was moderately strong. Hemoptysis (*n* = 130, 82.8%), chronic cough (*n* = 133, 84.7%), weight loss (*n* = 126, 80.3%), ongoing fatigue (*n* = 138, 87.9%), and shortness of breath (*n* = 111, 70.7%) were highly recognized by most respondents with mean scores all exceeding 0.71. Less commonly recognized symptoms were acute cough (*n* = 85, 54.1%), fever (*n* = 84, 53.5%), chest pain (*n* = 88, 56.1%), and chronic fever (*n* = 77, 49%), with mean scores ranging from 0.49 to 0.56. Finally, significant knowledge deficits were noted concerning rash (*n* = 14, 8.9%), severe headache (*n* = 26, 16.6%), and nausea (*n* = 27, 17.2%) as manifestations of TB (see Table 3).

#### 3.3.2. Mode and Control of Transmission

Knowledge of TB transmission was high, with most participants (*n*= 150, 95.5%) correctly identifying that TB is airborne and transmitted via sneezing or coughing (*M* = 0.96, *SD* = 0.21). Most respondents also recognized that TB is not transmitted through handshakes (*n* = 134, 85.4%), sharing dishes (*n* = 120, 76.4%), eating from the same plate (*n* = 117, 74.5%), or touching items in public places (*n* = 116, 73.9%), with means ranging from 0.74 to 0.85 (see Table 3). Participants also demonstrated a clear understanding that transmission of TB could be mitigated by covering the mouth when sneezing or coughing (*M* = 0.83; *SD* = 0.37). However, there appeared to be limited awareness (*n* = 60, 38.2%) that handwashing is not an effective preventive measure for TB (*M* = 0.38; *SD* = 0.49, see Table 3).

#### 3.3.3. Affected Demographics and Treatment

When asked who can be infected with TB, participants demonstrated a clear understanding that anyone can contract TB (*M* = 0.99; *SD* = 0.08) and that TB is curable (*M* = 0.87; *SD* = 0.34); however, not as many respondents understood the causative agents of TB (*M* = 0.66; *SD* = 0.47) and even fewer recognized that a TB infection is not the same as TB disease (*M* = 0.36; *SD* = 0.48, see Table 3).

Knowledge of the effectiveness of drugs provided by the health center as a cure was high (*M* = 0.89; *SD* = 0.32); however, very limited awareness (*n* = 46, 29.3%) was observed regarding directly observed therapy short course (DOTS) as a strategy used to treat TB (*M* = 0.29; *SD =* 0.46). Conversely, participants did demonstrate a strong understanding that non-medical approaches, such as herbal remedies (*n* = 156, 99.4%), home rest (*n* = 157, 100%), and praying (*n* = 153, 97.5%), are not appropriate cure regimens for TB (see Table 3).

### 3.4. TB-Related Attitudes of HCWs

Respondents displayed moderately positive attitudes toward TB-related practices (*M* = 3.87; *SD* = 0.37), with key findings highlighting HCWs’ positive attitudes toward adherence to protocols, commitment to safety measures, and a cautious approach to personal risk (see Table 4).

Respondents held the strongest positive attitudes towards adherence to infection control practices with high agreement on using face masks even if uncomfortable (*M* = 4.94; *SD* = 0.29), TB screening when symptomatic (*M* = 4.90; *SD* = 0.41), and maintaining uninterrupted treatment for patients who feel better (*M* = 4.64; *SD* = 0.74), underscoring the importance of continuous treatment. Participants also expressed high trust in laboratory results for sputum cultures (*M* = 4.64; *SD* = 0.75). The majority of respondents indicated they would not resign if assigned to a TB unit (*M* = 4.09; *SD* = 1.12) and expressed interest in attending TB seminars (*M* = 4.32; *SD* = 0.83) and learning more about TB (*M* = 4.25; *SD* = 0.85).

When asked about the need to isolate TB patients for treatment, there was less agreement (*M* = 3.93; *SD* = 1.15), as participants collectively expressed a cautious attitude toward allowing TB patients to leave the hospital soon after initiating treatment (*M* = 3.39; *SD* = 1.19). Attitudes toward initiating TB treatment for critically ill suspected cases before a confirmed diagnosis also showed more variability (*M* = 3.24; *SD* = 1.16).

Concern about personal risk was noted, with perceived susceptibility to TB (*M* = 4.15; *SD* = 0.95) and fear of contracting TB (*M* = 3.39; *SD* = 1.30) expressed by participants. Attitudes were also not as positive when asked if they would be willing to work in a TB unit (*M* = 3.53, *SD* = 1.34), and the idea of sharing utensils with infected family members was the most negatively viewed item (*M* = 2.10; *SD* = 1.32), reflecting significant discomfort (see Table 4).

### 3.5. Practices of HCWs

Eighty-four respondents (53.5%) reported prior experience working with TB patients and completed the portion of the questionnaire assessing their TB-related practices (Table 5). Overall, a strong TB-related practice among HCWs in Atlantic Canada was observed (*M* = 4.24, *SD* = 0.69), with a wide range noted (1.78, i.e., poor practice to 5.00, i.e., perfect practice).

#### 3.5.1. Patient Care

Most participants reported always performing hand hygiene and wearing personal protective equipment (PPE) before contact with TB patients or samples (*M* = 4.57; *SD* = 1.20), using N95 respirators during TB patient care/sample handling (*M* = 4.44; *SD* = 1.24), placing patients with TB disease in isolated rooms (*M* = 4.66; *SD* = 0.94), and separating TB patients from individuals with a known HIV seropositive status (*M* = 4.23; *SD* = 1.11, Table 5).

#### 3.5.2. Diagnostics

Diagnostic practices were more variable than patient care practices. There was high agreement on requesting sputum tests when TB was suspected (*M* = 4.45; *SD* = 0.84) and typically ensuring samples were sputum rather than saliva (*M* = 4.43 *SD* = 0.97). Ordering human immunodeficiency virus (HIV) tests when diagnosing TB disease was not as common (*M* = 3.22; *SD* = 1.36), and surprisingly, opening windows to increase natural ventilation in TB patient rooms had a much lower adherence rate (*M* = 2.83; *SD* = 1.53, see Table 5).

#### 3.5.3. Treatment

Other strong practices included avoiding the use of a wet or soiled N95 respirator (*M* = 4.79; *SD* = 0.82), requesting contact tracing for confirmed TB cases (*M* = 4.45; *SD* = 1.11), and ordering liver function tests before initiating anti-TB treatment (*M* = 4.32; *SD* = 0.91). Initiating prophylactic treatment for contacts testing positive for IGRA or the TST had a slightly lower mean score of 3.91 (*SD* = 1.01, see Table 5).

Unfortunately, due to the suboptimal reliability estimates, i.e., Cronbach’s alpha < 0.70, generated at the beginning of the analyses, the remainder of the planned analyses to examine the relationship between the knowledge, attitudes, and practices of HCWs in Atlantic Canada as well as with the aforementioned sociodemographic variables collected for this purpose was not conducted.

## 4. Discussion

This study sought to fill the gap in the scientific literature regarding TB-related knowledge, attitudes, and practices among HCWs in Atlantic Canada as well as identify targets for educational interventions to improve TB care and IPC. Evidence-based practice integrates scientific evidence, clinical expertise, and patient values to improve outcomes like mortality, patient safety, and quality of life, while reducing healthcare costs [9,22]. For HCWs to uphold EBP, continuous education and updates on health-related knowledge within their scope of practice are essential.

### 4.1. Knowledge Gaps

Our finding that most Atlantic Canadian HCWs have a moderate level of TB knowledge suggests preparedness to identify typical TB presentations for accurate diagnosis and timely intervention may be adequate [23]. However, knowledge of less specific symptoms, such as a common cough, fever, fever without a clear cause lasting more than seven days, and chest pain, was not widely recognized and is a cause for concern. Coupled with this gap, there appears to be a pervasive lack of recognition by HCWs of rare TB symptoms like rash, severe headache, and nausea, indicating clear targets for focused education initiatives. These gaps in recognizing atypical symptoms are important as they may delay diagnosis and treatment, potentially worsening patient outcomes [24].

These findings underscore the need for education to address HCW gaps in recognizing symptoms of infectious diseases. A systematic review and meta-analysis found that TB-focused training for HCWs, particularly emphasizing symptom recognition, significantly increased TB case detection rates, which can reduce diagnostic delays and enhance infectious disease control [23]. Thus, educational initiatives emphasizing both common and less common symptoms could enhance patient care and improve public health outcomes.

Although participants demonstrated strong overall knowledge of effective TB treatment options, they showed limited awareness of DOT, a key WHO-recommended strategy to ensure treatment adherence and tolerability [25]. This is disconcerting and may be due to a variety of reasons, including inconvenience, insufficient training, ineffective communication in the workplace where up-to-date guidelines are disseminated, or the presence of stigma surrounding TB [26]. Enhancing HCWs’ understanding of TB treatment strategies, particularly among physicians and nurse practitioners, will be crucial, as limited awareness could result in failures or delays in initiating necessary treatment for TB patients, compromising outcomes and public health efforts.

Participants demonstrated moderately strong baseline knowledge of TB’s primary pathogen but lacked an understanding of the distinction between TB infection (latent TB infection: a dormant, asymptomatic, non-contagious stage) and TB disease (active TB disease: a symptomatic, contagious stage). Education on these phases is essential as misunderstanding may lead to the unnecessary treatment of TB infection or the delayed treatment of TB disease, increasing transmission risks and adverse outcomes [27,28].

Knowledge regarding TB transmission was strong among participants, with most correctly identifying true modes of transmission and rejecting false ones (e.g., sharing plates, handshakes). This suggests HCWs understand TB transmission pathways, supporting safe practice in line with EBP. Similarly, participants demonstrated strong knowledge of TB transmission control, correctly identifying effective measures such as covering the mouth when sneezing or coughing while rejecting non-EBPs like avoiding handshakes, closing windows, or relying on good nutrition and prayer.

However, knowledge gaps were noted in some areas of transmission control. Only 66% and 38% of participants correctly recognized that avoiding shared dishes and washing hands after touching public items, respectively, are ineffective transmission control strategies. Such misconceptions can perpetuate existing stigma around TB by reinforcing misinformed fears of casual transmission. Stigma in healthcare settings is well-documented, manifesting as denial of care, sub-standard treatment, prolonged wait times, and mistreatment [29,30]. Educational interventions in healthcare settings and the dissemination of evidence-based knowledge have been shown to reduce stigmatizing attitudes among HCWs. A recent systematic review reported significant reductions in stigma and discriminatory behaviors following educational interventions, highlighting the importance of ongoing training [29].

### 4.2. Attitudes

In addition to knowledge gaps, this study found moderately positive HCW attitudes toward TB-related practices. Participants showed strong willingness to be screened for TB, work with TB patients, and wear face masks while attending to TB patients. These findings align with Shrestha et al. [31], who reported that most Nepali HCWs had positive attitudes toward TB infection control, particularly respiratory protection. Dil et al. [32] further support this, showing a direct correlation between attitude and practice in HCWs during the COVID-19 pandemic, further supporting the link between attitude and practice in healthcare settings. However, HCWs expressed negative attitudes toward sharing cutlery, plates, and glasses with individuals infected with TB, mirroring transmission control knowledge gaps. These results suggest that knowledge gaps may contribute to negative attitudes, which can ultimately influence practices [32]. Negative attitudes, such as implicit racial or ethnic biases, can lead to inequitable care, hinder patient–provider interactions, and harm patient outcomes [33]. In Canada, TB-affected populations (e.g., Indigenous and immigrant populations) face heightened risks of discrimination due to implicit bias [34].

Conversely, strong positive attitudes were also evident, as HCWs expressed a willingness to attend TB training. This suggests an opportunity to address knowledge gaps and negative attitudes, promoting equitable care. TB training has been shown to increase TB case detection [23]. While various intervention modes have been used for continuing educational programs, the literature suggests that interactive, high-fidelity simulation-based education, rather than traditional didactic teaching methods, can improve patient safety, increase trainees’ confidence, and reduce medical errors [35]. Such approaches not only build knowledge but also address biases and misconceptions, ultimately promoting equitable care.

### 4.3. Practice

Interestingly, despite the low TB incidence in Atlantic Canada in 2021 (1–1.7/100,000 population) [5], HCWs in this region self-reported strong TB-related practice, including hand hygiene, PPE use, N95 mask handling, isolating TB patients, sputum testing, and contact tracing, aligning with EBP. Han [36] found an association between handwashing and reduced TB incidence and mortality, emphasizing its role in TB prevention.

Clear gaps in practice were identified with inconsistent practices identified in natural ventilation strategies, initiating anti-TB treatment prior to laboratory confirmation, and ordering HIV tests for TB patients. These findings align with a prior study that reported suboptimal practices in treatment initiation and ventilation strategies for TB patients. Our findings highlight the need for further education to optimize TB-related practices [14]. Of particular interest was how poorly recognized natural ventilation was as a best practice. This is interesting as it has a strong, well-established evidence base for positive impacts when managing infectious disease [37]. Perhaps in the wake of the COVID-19 pandemic, where national and provincial safety protocols emphasized vaccination, physical distancing, avoiding crowds, and wearing masks [38], HCWs have drifted from those best practices, i.e., natural ventilation, which were not emphasized during this time. This is interesting and deserves further investigation if other best practices that were not central to the COVID-19 response have been “forgotten”; whatever the cause for this lack of awareness, there should be an immediate and clarion call to HCWs, and perhaps the general public, reminding them of the advantages of natural ventilation when managing infectious disease. In addition, the limited recognition of natural ventilation as a best practice for preventing TB transmission among HCWs in Atlantic Canada may also be attributed to the fact that most (*n* = 119, 76%) of the respondents in our study work in hospitals where centralized mechanical ventilation systems are commonly used. This context likely influences their perceptions and familiarity with natural ventilation strategies, potentially leading to an underappreciation of its effectiveness in appropriate settings.

### 4.4. Limitations

This study has several limitations. First, TB-related practices were self-reported, which may not accurately reflect actual practices and likely represent the views of HCWs particularly concerned about TB who chose to participate, introducing potential self-selection bias. Future studies should incorporate objective assessments, such as naturalistic observations. Further, it should be noted that those HCWs who did not have experience caring for a TB patient did not answer these TB-related practice questions in this study; therefore, it is unknown what the practice level would be for those uninitiated HCWs in the region if assigned to care for a patient with TB. Additionally, the results of the current study may not be generalized across all four Atlantic provinces due to the gross overrepresentation of respondents from NB. Broader sampling from the other Atlantic provinces is warranted in future research.

Secondly, while this study’s questionnaire was informed by WHO’s KAP template and other anchoring articles, the psychometric properties of the tools were suboptimal. The KAP subscales all demonstrated Cronbach’s alpha values below the commonly accepted threshold of 0.70, indicating low internal consistency. This limitation may reduce the reliability and interpretability of the findings and impact the generalizability of the results. While this study provides valuable preliminary insight into the KAP of HCWs in Atlantic Canada, further refinement and formal validation of the survey instrument are needed before wider application in future research. Furthermore, due to the low internal consistency of the KAP subscales, subgroup analyses (e.g., by healthcare profession, prior TB-specific training, or TB-related experiences) were not conducted. Future research using improved survey instruments may allow for meaningful stratified analyses.

Finally, participant incentives might have introduced bias, limiting sample representativeness. As an online survey, participation may have been influenced by digital literacy or access, potentially excluding some HCWs from participating in this study.

### 4.5. Summary of Targets for Educational Interventions

Given the significant knowledge gaps identified among HCWs in Atlantic Canada, targeted educational interventions in nursing schools, community colleges, medical schools, hospitals, and long-term care facilities are needed to improve the early detection, diagnosis, and management of TB cases that are expected to increase within the region [39]. Firstly, in-service training should emphasize both common and atypical TB symptoms, including (1) persistent fever without a clear cause, (2) chest pain, (3) rash, (4) severe headache, and (5) nausea, to reduce delays in diagnosis and prevent progression to severe disease. Secondly, misconceptions around TB transmission, such as the belief that avoiding shared dishes or washing hands after touching public items are effective control measures, must be addressed to improve effective prevention strategies and reduce TB-related stigma. Thirdly, practical competencies should also be strengthened, including (1) the increased use of DOT, (2) understanding the critical distinction between TB (latent) infection and (active) TB disease, and (3) the role of natural ventilation in infection control. Lastly, it is recommended that a program grounded at one of the local Atlantic universities is established to further this work and ensure that knowledge translation and targeted training resources are delivered to healthcare facilities and workers in a timely manner to enhance TB control efforts and improve patient outcomes amidst the rapidly shifting demographics of Atlantic Canada.

## 5. Conclusions

This study revealed encouraging results but also uncovered significant gaps in TB-related KAPs among HCWs in Atlantic Canada. These findings underscore the need for targeted education and training initiatives to strengthen TB preparedness and promote equitable care for patients in the region.

## Figures and Tables

**Table 1 tropicalmed-10-00214-t001:** Sociodemographic characteristics of HCWs in Atlantic Canada (*N* = 157).

Characteristics	Category	Count
Age	19–29	19 (12%)
	30–39	44 (28%)
	40–49	38 (24%)
	50–59	42 (27%)
	60–69	14 (9%)
Sex	Male	11 (7%)
	Female	145 (92.4%)
	Unknown	1(0.6%)
Ethnicity	Black/African	4 (2.5%)
	White	139 (88.5%)
	South Asian (e.g., Indian or Pakistani)	2 (1.3%)
	Southeast Asian (e.g., Filipino or Thai)	8 (5.1%)
	Indigenous	2 (1.3%)
	Other (e.g., mixed)	2 (1.3%)
Were you born in Canada?	Yes	134 (85%)
	No (e.g., Brazil, Libya, Philippines)	23 (15%)
Have you lived outside of Canada?	Yes (e.g., Peru, United Arab Emirates)	39 (24.8%)
	No	115 (73.3%)
	Unknown	3 (1.9%)
Do you currently live in Canada?	Yes	156 (99.4%)
	No	1 (0.6%)
Province of residence	New Brunswick	122 (77.7%)
	Nova Scotia	21 (13.4%)
	Prince Edwards Islands	1 (0.6%)
	Alberta	2 (1.3%)
	Ontario	1 (0.6%)
	Not specified	10 (6.4%)

**Table 2 tropicalmed-10-00214-t002:** Professional demographic characteristics (*N* = 157).

Variable	Category	Count
Level of training *	College	89
	University	62
Employment status	Full-time	106
	Part-Time	51
Employment role	Registered Nurse	105
	Licensed Practical Nurse	18
	Personal Support Worker	1
	Registered Respiratory Therapist	18
	Supervisor/Admin/Educator/ Low-Contact Personnel	13
	Occupational Therapist	2
Place of employment *	Hospital	119
	Long-Term Care	16
	Community/Clinic	11
Current province of practice	New Brunswick	132
	Nova Scotia	21
	Prince Edwards Island	1
	Newfoundland and Labrador	1
	Multiple Provinces	2
Previous provinces of practice *	Always Worked in Current Province	80
	New Brunswick	2
	Nova Scotia	9
	Prince Edwards Island	0
	Newfoundland and Labrador	0
	Alberta	3
	Ontario	17
	Manitoba	3
	Quebec	2
	British Columbia	1
	Saskatchewan	1
	2+ multiple	13
Previous countries of practice *	Canada	122
	Outside Canada	23
Previous work experience in high TB setting	Yes	20
	No	125
Have you received any tuberculosis specific training in the past?	Yes	60
	No	97
Have you personally suffered from tuberculosis?	Yes	5
	No	152
Do you know anyone that has or has had tuberculosis?	Yes	66
	No	91

* Respondents chose not to respond to some questions therefore numbers do not add up to *N* = 157.

**Table 3 tropicalmed-10-00214-t003:** Knowledge of HCWs about TB.

Variable	Mean	Std. Deviation
Symptoms		
Rash	0.09	0.29
Cough	0.54	0.50
Cough lasting 3+ weeks	0.85	0.36
Hemoptysis	0.83	0.38
Severe headache	0.17	0.37
Nausea	0.17	0.38
Weight loss	0.80	0.40
Fever	0.54	0.50
Fever without clear cause that lasts more than 7 days	0.49	0.50
Chest pain	0.56	0.50
Shortness of breath	0.71	0.46
Ongoing fatigue	0.88	0.33
Do not know	0.05	0.22
Others (please explain)	0.08	0.28
Mode of transmission		
Through handshakes	0.85	0.36
Through the air when a person with TB coughs or sneezes	0.96	0.21
Through sharing dishes	0.76	0.43
Through eating from the same plate	0.75	0.44
Through touching items in public places (doorknobs, handles in transportation, etc.)	0.74	0.44
Transmission control		
Avoid shaking hands	0.87	0.34
Covering mouth and nose when coughing or sneezing	0.83	0.37
Avoid sharing dishes	0.66	0.47
Washing hands after touching items in public places	0.38	0.49
Closing windows at home	0.99	0.08
Through good nutrition	0.78	0.41
By praying	0.98	0.16
Affected demographics		
Anybody	0.99	0.08
Only economically disadvantaged individuals	0.98	0.14
Only unhoused individuals	0.99	0.08
Only individuals with alcohol use disorder	0.00	0.00
Only drug users	0.00	0.00
Only people living with HIV/AIDS	0.99	0.11
Only individuals who have been incarcerated	0.99	0.08
Treatment strategy		
Herbal remedies	0.99	0.08
Home rest without medicine	0.00	0.00
Praying	0.98	0.16
Specific drugs given by health center	0.89	0.32
Directly Observed Therapy (DOT)	0.29	0.46
Tuberculosis is caused by **_______**	0.66	0.47
Tuberculosis infection is the same as tuberculosis disease	0.36	0.48
Can tuberculosis be cured?	0.87	0.34
Knowledge Total	29.34	3.25

**Table 4 tropicalmed-10-00214-t004:** Descriptive statistics for TB-related attitudes.

Variable	Mean	Std. Deviation
Would you be willing to work in a tuberculosis unit?	3.453	1.34
Would you resign from work if you are posted to a tuberculosis unit? *	4.09	1.12
Would you be willing to be screened for tuberculosis if you had suggestive symptoms?	4.90	0.41
Do you think all tuberculosis patients should be isolated for treatment?	3.93	1.15
Is it ok to allow a tuberculosis patient to leave the hospital soon after initiating appropriate treatment?	3.39	1.19
Would you be willing to attend seminars on tuberculosis?	4.32	0.83
Would you recommend the suspension of treatment if a tuberculosis patient is feeling better? *	4.64	0.74
Would you start tuberculosis treatment for a tuberculosis patient before diagnosis is confirmed if a suspected tuberculosis patient is very ill?	3.24	1.16
Would you use a face mask when dealing with a tuberculosis patient even when it is uncomfortable?	4.94	0.29
Would you trust the result provided by the laboratory on sputum cultures?	4.64	0.75
I would not accept to examine/treat a tuberculosis patient *	4.47	0.94
Do you think you could get tuberculosis?	4.15	0.95
Are you scared of getting tuberculosis? *	3.39	1.30
Would you continue to socialize with your friends if they were diagnosed with tuberculosis?	3.34	1.34
Would you share the same cutlery, plates, and glasses with a family member if they were infected with tuberculosis?	2.10	1.32
Would you say that tuberculosis is a stigmatized disease?	3.71	1.19
Would you like to learn more about tuberculosis?	4.25	0.85
Do you feel tuberculosis is a major threat to public health in Canada?	3.28	1.13
Do you think there is a need for improvement in tuberculosis control in your region?	3.21	1.04
Attitude Mean	3.87	0.37

* items reverse-scored for increased ease in interpreting data.

**Table 5 tropicalmed-10-00214-t005:** Descriptive statistics for TB-related practice.

Variable	N	Mean	Std. Deviation
I usually perform hand hygiene and wear personal protective equipment (PPE) before contact with tuberculosis patients/samples	79	4.57	1.20
I usually wear an N95 respirator when caring for patients with tuberculosis or working on tuberculosis samples	75	4.44	1.24
I request sputum tests when I suspect active tuberculosis	58	4.45	0.84
I always put patients with active tuberculosis in isolated rooms	68	4.66	0.94
I open windows when possible, in tuberculosis patients’ rooms to increase natural ventilation	48	2.83	1.53
I order an HIV test when I diagnose active tuberculosis	50	3.22	1.36
I always separate patients with known tuberculosis from HIV patients	52	4.23	1.11
Using a wet or soiled N95 respirator is unacceptable	81	4.79	0.82
I always make sure that samples are sputum and not saliva before sending them to the lab or before testing in the lab	65	4.43	0.97
I commence anti-tuberculosis drugs on suspected tuberculosis cases before laboratory confirmation	41	2.80	1.25
I request contact tracing for all confirmed tuberculosis cases	55	4.45	1.12
I request liver function tests before starting anti-tuberculosis treatment	47	4.32	0.91
I initiate prophylaxis treatment for the contacts of active tuberculosis cases who test positive for interferon-gamma release assay (IGRA)/tuberculin skin test (TST)	44	3.91	1.01
Practice Average	81	4.24	0.68

## Data Availability

The data presented in this study are openly available in Open Science Framework at https://doi.org/10.17605/OSF.IO/ZNY36.

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
