# Peer review of "Tuberculosis-Related Knowledge, Attitudes, and Practices Among Healthcare Workers in Atlantic Canada: A Descriptive Study"

_tropicalmed, 2025, doi:10.3390/tropicalmed10080214_

Round 1
Reviewer 1 Report
Comments and Suggestions for Authors
The article by Oh et al presents a well-structured study examining the knowledge, attitudes, and practices regarding tuberculosis (TB) among healthcare workers (HCWs) in Atlantic Canada—which has limited data on the same. The study is methodologically sound, with a clear rationale. The study has a public health relevance, especially focussed on regional disparities in TB knowledge and preparedness. The authors noted good practices among HCWs, especially regarding transmission, protective gear use, and attitude towards engaging in TB care; besides, lack of awareness towards atypical TB symptoms (rash, nausea), DOTS and the distinction between latent infection and active disease, and inconsistent practices in ventilation and pre-treatment protocols were documented. The study provides valuable insights for developing targeted training in low-incidence TB settings. I suggest MINOR revisions prior to recommending the study for publication:
- Assert the psychometric limitations directly in the Discussion (currently scattered).
- Explain abbreviations (e.g., DOTS, IGRA, TST) upon first use in the abstract or introduction.
- The methodology does not indicate KAP scores by profession/ experience with TB/ prior training; some scales reported Cronbach’s α < 0.70-- the authors can clarify
- Address limitations of online surveys beyond just incentive bias—such as., digital literacy bias; address overrepresentation of HCWs from New Brunswick
- Rephrase generic terms, like "strong practice" or “positive attitude,” to replace with quantitative indicators where available.
- Give a brief discussion on future interventions, strategies, recommendations. The authors can refer to (https://doi.org/10.3390/microorganisms13010178) to address the same
Reviewer 2 Report
Comments and Suggestions for Authors
Tuberculosis Related Knowledge, Attitudes, and Practices Among Healthcare Workers in Atlantic Canada: A Descriptive Study
Dear authors:
Thank you for submitting your article to “Tropical Medicine and Infectious Disease”. The study explores the knowledge, attitudes, and practices of Atlantic Canadian healthcare workers and identifies opportunities for targeted educational interventions to enhance tuberculosis care and infection prevention and control.
The study conducted in Atlantic Canada shows a perspective on the intended research. With a markedly subjective intention, it includes local information, which could be useful in understanding the (not well-described) population. However, the article is very basic in its approach, rigor, depth of study, use of statistical/epidemiological indicators, etc., making it more appropriate for research of a narrow scientific nature. The presentation of results is based primarily on absolute and relative frequencies, which denotes the elementary content and, under better judgment, the lack of alignment with the quality of a prestigious journal.
Reviewer 3 Report
Comments and Suggestions for Authors
-
Line 12–13: The abstract should specify the gap in existing literature more clearly and provide a concise justification for the study. Consider referencing a specific lack of regional KAP data for HCWs.
-
Line 15–19: While the data collection method is stated, the survey instrument description is limited. It would be helpful to elaborate on the validity/reliability testing (e.g., Cronbach’s alpha values).
-
Line 23–25: Specify the cutoffs or scoring scale interpretation for “moderately high knowledge” and “positive attitudes” to enhance clarity.
-
Line 66–67: The rationale for focusing specifically on Atlantic Canada would benefit from elaboration—perhaps mention differences in healthcare delivery or TB burden by province.
-
Line 74: The sentence is grammatically incorrect: “was secured for the planned data analysis..” — please revise for clarity and punctuation.
-
Line 78–82: The adaptation process of the WHO survey tool could be described in more detail, including specific items added or changed.
-
Line 115–117: It would strengthen ethical transparency to state whether participants had the option to withdraw and whether IP addresses were logged or anonymized.
-
Line 164–169: Consider organizing symptom knowledge results into “highly recognized,” “moderately recognized,” and “poorly recognized” categories for interpretive clarity.
-
Line 185–190: The poor knowledge of DOTS and latent vs. active TB indicates an educational gap. A discussion of why this knowledge might be lacking (e.g., training gaps, system-level emphasis) would be helpful.
-
Line 236: A mean score of 2.83 for natural ventilation is concerning; this should be discussed more explicitly in the Discussion section as a critical practice gap.
-
Line 295–317: The paragraph would benefit from clearer topic sentences and subheadings to distinguish between attitude findings and implications for bias/stigma.
-
Line 341: The limitations around psychometric strength should be more clearly acknowledged earlier in the Methods or Results section as well.
-
General: Several tables are embedded in the text without clear figure/table references in the Results narrative. For example, Table 3 is presented but not clearly referenced in the surrounding discussion. Add explicit citations to each table in the relevant text sections.
Sentence structure and phrasing should be reviewed for consistency and academic tone. For instance, “strong positive attitudes were also evident…” could be revised to a more objective tone, e.g., “Participants also reported favorable attitudes regarding…”
Reviewer 4 Report
Comments and Suggestions for Authors
The manuscript presents locally relevant data on a topic of importance for global health. The study addresses a gap in knowledge regarding TB KAP among healthcare workers (HCWs) in Atlantic Canada, a low-incidence setting. The findings have implications for targeted educational interventions to improve TB care and infection control practices.
As most of the analysis has been done using statistical work, the following are the queries and observations about statistical aspects:
- All the ranges show obvious min and max values of 0 to 1 or 1 to 5, showing slight variation, so these columns may be deleted for better data presentation.
- The authors have provided only exploratory data analysis; however, no confirmatory data analysis shows whether Socio-Demographic Characteristics or professional Demographic Characteristics affect knowledge and practices. Authors can refer to the article mentioned below.
- Knowledge, attitudes, and practices (KAP) toward COVID-19: a cross-sectional study in South Korea Minjung Lee, Bee-Ah Kang and Myoungsoon You
- No mention of how many participants actually replied correctly to the knowledge questions
- What could be the correlation coefficient value between knowledge scores, attitude scores & practice scores?
- I wonder what the use of doing KAP is in a low-prevalence and highly developed country?
- The application of Snowball sampling or a non-probability sampling technique is somewhat unclear, as it is useful for studying populations that are difficult to access or identify using traditional methods. HCW can’t be categorised as hard to reach.
Other observations and comments:
- The overrepresentation of respondents from New Brunswick needs to be addressed. Discuss potential biases due to this imbalance and acknowledge limitations in generalizing findings to all of Atlantic Canada.
- Address the potential for selection bias due to the survey's voluntary nature. Those more interested in TB might be overrepresented.
- The authors acknowledge that the questionnaire's reliability and validity remain unclear, and Cronbach's alpha values are below the recommended cut-off of 0.70. It is important to acknowledge that the survey instrument's psychometric properties were not strong. The authors should discuss this limitation and its potential impact on the generalizability of the findings.
- The adaptation of the WHO survey guidelines is a good starting point, but how were the modifications made to tailor it to the Atlantic Canadian context? Was expert input sought? What specific changes were implemented, and what was the rationale behind them?
- The pilot testing on five HCWs is minimal. What specific feedback was received during pilot testing, and how did this feedback influence the final version of the questionnaire? A more detailed discussion of the pilot testing process is needed.
- While the manuscript mentions the need for targeted educational interventions, it could provide more specific and actionable recommendations. What specific topics should be included in educational programs based on the identified knowledge gaps and inconsistent practices? What delivery methods would be most effective for reaching HCWs in Atlantic Canada?
Round 2
Reviewer 1 Report
Comments and Suggestions for Authors
None
Reviewer 4 Report
Comments and Suggestions for Authors
The authors have sufficiently responded and modified the manuscript accordingly...a few things that are incorporated in response may also be discussed in manuscript as like the response to point no. 7. Other things are accepted.